# Association of ACE2 Gene Variants with the Severity of COVID-19 Disease—A Prospective Observational Study

**DOI:** 10.3390/ijerph191912622

**Published:** 2022-10-02

**Authors:** Jerzy Sienko, Izabela Marczak, Maciej Kotowski, Anna Bogacz, Karol Tejchman, Magdalena Sienko, Katarzyna Kotfis

**Affiliations:** 1Department of General Surgery and Transplantology, Pomeranian Medical University, 70-111 Szczecin, Poland; 2Department of Neurology, SUNY Downstate Medical Center, Brooklyn, NY 11023, USA; 3Department of Stem Cells and Regenerative Medicine, Institute of Natural Fibers and Medicinal Plants, 62-064 Plewiska, Poland; 4Department of Pediatrics, Endocrinology, Diabetology, Metabolic Diseases, and Cardiology, Pomeranian Medical University, 71-252 Szczecin, Poland; 5Department of Anesthesiology, Intensive Therapy and Acute Intoxications, Pomeranian Medical University, 70-111 Szczecin, Poland

**Keywords:** SARS-CoV-2, coronavirus, angiotensin receptors, gene polymorphism, single nucleotide polymorphism

## Abstract

The COVID-19 pandemic, caused by severe acute respiratory syndrome coronavirus 2 virus (SARS-CoV-2), has triggered an enormous scientific response. Many studies have focused on understanding the entry of the SARS-CoV-2 virus into the host cell. The angiotensin-converting enzyme-2 (ACE2) is recognized as the host receptor used by SARS-CoV-2 to enter its target cells. Recent studies suggest that ACE2 gene polymorphisms might be candidates for genetic susceptibility to SARS-CoV-2 infection. The aim of this study is to evaluate the influence of ACE2 polymorphisms on COVID-19 disease risk and severity. In our study, we confirmed that there is a statistically significant increased risk of a more severe disease course of SARS-CoV-2 infection associated with the need for hospitalization in intensive care for patients with specific polymorphisms of the ACE2 gene. The most significant correlation was found for variant ACE2 rs2285666 (AA allele, OR = 2.12, *p* = 0.0189) and ACE2 rs2074192 (TT allele, OR = 2.05, *p* = 0.0016), and for ACE2 rs4646174 (GG allele, OR = 1.93, *p* = 0.0016), ACE2 rs4646156 (TT allele OR = 1.71, *p* = 0.008) and ACE2 rs2158083 (TT allele OR = 1.84, *p* = 0.0025). In conclusion, our findings identify that certain ACE2 polymorphisms impact the severity of COVID-19 disease independently of other well-known risk factors.

## 1. Introduction

Severe acute respiratory syndrome virus 2 (SARS-CoV-2) has caused a global pandemic of respiratory infection, called Coronavirus disease 2019 (COVID-19). The first cases of SARS-CoV-2 infection were reported in December 2019 in Wuhan Province of China, however, the region where the virus originated is still debated. In a few months, the coronavirus spread rapidly to other parts of the world [1]. Based on the data from a COVID-19 Dashboard, created by the Center for Systems Science and Engineering (CSSE) at Johns Hopkins University, at the time of writing this manuscript, there were over 244 million confirmed cases of SARS-CoV-2 infections and nearly 5 million deaths around the word. SARS-CoV-2 belongs to a family of Coronaviruses, having relatively large, enveloped viruses with non-segmental, positive-stranded RNA molecules. Most coronaviruses, such as 229E, NL63, HKU1, and OC43, typically cause mild respiratory infection [2,3]. However, past decades have brought us three highly lethal members of coronavirus, including SARS-CoV-1, MERS-CoV, and novel SARS-CoV-2, that cause Severe Acute Respiratory Syndrome (SARS), Middle East Respiratory Syndrome (MERS), and Coronavirus disease 19 (COVID-19), respectively. In contrast, SARS-CoV-2 is much more contagious than SARS-CoV-1 or MERS-CoV [4,5]. However, SARS-CoV-1 and MERS-CoV have higher mortality rates than novel SARSCoV-2 [6]. Coronaviruses derived their name from the Latin word “crown”, which refers to the crown-like appearance of the virus when seen under an electron microscope [7]. SARS-CoV-2 has four structural proteins: membrane (M), envelope (E), spike (S), and nucleocapsid (N) proteins. Like SARS-CoV-1, the surface spike protein of SARS-CoV-2 is essential for viral attachment and entry into the host cell via the angiotensin-converting enzyme 2 (ACE2) receptor [8,9]. The ACE2 receptor is highly expressed in capillary-rich organs, including the lungs, which may explain the predominance of respiratory symptoms of COVID-19 [10,11,12]. The ACE2 receptor is a crucial element of the renin-angiotensin system (RAS) responsible for regulating blood pressure. Its catalytic domain has over 40% identity to its homolog angiotensin-converting enzyme (ACE) [13]. The ACE is located in the lungs and converts inactive angiotensin I (AngI) to active angiotensin II (AngII) [14]. Angiotensin II binds to the AT1 receptor, causes vasoconstriction, and promotes inflammation or might enhance thrombosis via the AT4 receptor [15,16]. In contrast, the ACE2 enzyme catalyzes the hydrolysis of angiotensin II to inactive angiotensin (1-7) (Ang (1-7)), which is a vasodilator agent with antihypertensive and anti-proliferation properties [16,17]. Thus, ACE2 is an important RAS negative factor that balances the ATII/Ang(1-7) ratio. The SARS-CoV-2 virus interacts with the ACE2 receptor, causing lower ACE2 availability and decreased breakdown of Ang II. Down-regulation of the ACE2 receptor might lead to severe acute respiratory failure with an increased risk of cardiac injury and thrombosis [18,19]. The ACE2 gene is located on chromosome Xp22 and is a polymorphic gene with about 140 single nucleotide polymorphisms (SNPs). Many studies confirm that ACE2 polymorphisms are associated with a risk of primary hypertension and other cardiovascular diseases [20,21]. The ACE2 variants associated with hypertension include rs2285666, rs879922, rs4646188, rs2106809, rs4240157, rs4830542, rs2158083, rs879922, rs1514283, rs2074192, rs4646155, rs4646176, rs4646174 and rs233575. Meanwhile, left ventricular hypertrophy ACE2 polymorphisms include rs2106809, rs2074192, rs4646156, rs879922, rs4240157 and rs233575 [22]. In the study, we selected five single nucleotide polymorphisms (SNPs) (rs2074192, rs2158083, rs2285666, rs4646156, rs4646174) based on the frequencies of the variants in the European population. Allele frequencies of these polymorphisms in the European population are presented in Table 1. Moreover, several studies have indicated that those SNPs might be involved in SARS-CoV-2 infection [23,24]. The rs228666, located in intron 3, may change the mRNA splicing and affect the expression of the ACE2 gene, causing greater affinity to a novel coronavirus. Similar properties were found for rs2158083 intronic variant [25,26]. The rs2074192, an intronic variant, was found to induce RNA changes of secondary structure that may cause ACE2 transcription/translation imbalance. This dysregulation changes SARS-CoV-2 binding to angiotensin receptors [26]. The rs4646156 and rs4646174 were highly associated with left ventricular hypertrophy and primary hypertension, respectively. Cardiovascular diseases are well-known common co-morbidities of SARS-CoV-2 infection that increase the risk of hospitalization and death [27,28]. The aim of the present study was to evaluate the influence of ACE2 gene polymorphisms (rs2074192, rs2158083, rs2285666, rs4646156, rs4646174) on COVID-19 disease risk and severity in the Polish population, and to understand the genetic causes of susceptibility for SARS-CoV-2 infection.

## 2. Materials and Methods

In this prospective observational study, we enrolled adult patients of both sexes, who were positive for the SARS-CoV-2 virus, confirmed with PCR standardized testing, who were symptomatic, and who were not vaccinated. The study involved 188 individuals that were classified according to the severity of the infection: Group I with no or mild symptoms, such as fever, cough, muscle aches, loss of pain/taste, who were treated at home or in isolation treatment places; Group II with a severe course of coronavirus disease that required hospitalization in the intensive care unit and mechanical ventilation. To minimize the possible interferences, such as older age, severe obesity, or severe comorbidities, on the results we optimize group II via exclusion criteria for study subjects. The exclusion criteria were as follows:The average post-retirement age (over 68 years).Obesity class 2 (BMI > 35 kg/m^2^) except for relatively younger patients or patients without severe chronic illness.Chronic kidney disease (CKD) treated with dialysis.Unknown (patient with SBP above 140 mmHg or DBP above 90 mmHg in at least two different measurements with no previous history of hypertension) or decompensated (hypertension crisis SBP above 180 mmHg and/or DBP above 120 mmHg) hypertension.Unknown (patient with random glucose level above 200 mg/dL with no previous history of diabetes) or decompensated (diabetic ketoacidosis on admission) diabetes mellitus.

According to the manufacturer’s protocol, genomic DNA was isolated from peripheral blood leukocytes using a commercially available PureLink™ Genomic DNA Mini Kit (Thermo Fisher Scientific, Waltham, MA, USA). Briefly, 200 µL of each blood sample was digested using proteinase K and RNase for 10 min at 55 °C. Afterward, the lysis buffer and ethanol were added, and the samples were loaded onto a silica-based membrane. After washing and elution steps, the DNA concentration was measured using Epoch Microplate Spectrophotometer (BioTek, Winooski, VT, USA). Analysis of the ACE2 (rs2074192, rs2158083, rs2285666, rs4646156, rs4646174) polymorphisms was performed by real-time PCR method using LightCycler^®^ 96 system (Roche Diagnostics, Basel, Switzerland). Sets of LightSNiPs (TibMol-biol, Berlin, Germany) to determine ACE2 polymorphisms contained appropriate concentrations of specific primers and probes for the amplified fragment and were prepared according to the manufacturer’s instructions. The PCR program was initiated at 95 °C for 10 min. Each PCR cycle comprised a denaturation step at 95 °C for 10 s, an annealing step at 60 °C for 10 s, and an elongation step at 72 °C for 15 s (45 cycles). The final stage was the melting of products because of temperature rise to 95 °C. The reaction composition of a single sample was as follows: H_2_O—6.7 μL, LightSNiP—0.5 μL, LightCycler480 Genotyping Master—1 μL, MgCl_2_ (25 mM)—0.8 μL, DNA (50 ng)—1 μL. The analysis of the genotyping was based on the melting curve using LightCycler^®^ 96 Basic Software (Roche Diagnostics, Basel, Switzerland).

Statistical analysis was performed on raw data using Statistica 13 software package (TIBCO Software Inc., 3307 Hillview Avenue Palo Alto, CA, USA). Polymorphisms within study groups were analyzed using frequency tables and Pearson’s chi-squared, as well as nonparametric R Spearman, rang tests. Odds ratios and confidence intervals for polymorphisms and concomitant diseases were calculated using the creator of logistic regression. They all were analyzed as one-factor risk evaluation and in conjunction as a multifactor set. Continuous variables were analyzed with the use of Student *t*-test and Kruskal–Wallis ANOVA rang test. Statistical tests were chosen according to the input data and the cause–effect analytic questions. Statistical significance was set at *p* < 0.05.

All experimental protocols were approved by the Ethical Committee of the Pomeranian Medical University in Szczecin (Approval number: KB-0012/88/2020; approval date, 22 June 2020). The study was conducted in accordance with the Helsinki Declaration (1975, revised 2000). We obtained approval from the Ethical Committee of the Pomeranian Medical University in Szczecin to perform our study without written informed consent from all study subjects, due to the observational nature of the study (Approval number KB-0012/88/2020; approval date, 22 March 2021).

## 3. Results

The study involved 188 individuals that were positive for the SARS-CoV-2 virus between September 2020 and May 2021. Patients were classified according to the severity of the infection: Group I (101 patients) with no or mild symptoms, such as fever, cough, muscle aches, loss of smell, or loss of taste, who were treated at home or in isolation treatment places; and Group II (87 patients) with a severe course of coronavirus disease that required hospitalization in the intensive care unit and mechanical ventilation. The general demographic and clinical characteristics of groups I and II are present in Table 2. In Group II the percentage of men was higher (72.41%, *p* < 0.001), as was the BMI (*p* = 0.001). Additionally, there was a strong statistical significance between type 2 diabetes mellitus and severe SARS-CoV-2 infection outcome (OR = 3.072, *p* = 0.0428). Odds ratios and confidence intervals for polymorphisms and concomitant diseases were calculated using logistic regression. They all were analyzed as one-factor risk evaluation and in conjunction as a multifactor set. Diabetes and ACE2 polymorphisms are independent factors for the severity of COVID-19. However, there was no statistical significance between ACE2 gene polymorphisms or diabetes and the severity of COVID-19. Diabetes and ACE2 gene variants seem to be two independent factors of the novel coronavirus outcomes.

The odds ratio (OR) of being in Group II was 7.509 greater for patients with ischemic heart disease, but there was no statistical significance (*p* > 0.05 and *p* < 0.1). Interestingly, hypothyroidism was associated with low significance to a milder course of COVID-19 disease (OR = 0.208, *p* = 0.006).

### 3.1. Association of Genetic Variants of ACE2 Gene with the Risk of Severe COVID-19 Disease

In our study, we confirmed that there is a statistically significant increased risk of a more severe disease outcome of SARS-CoV-2 infection associated with hospitalization in the intensive care unit for specific polymorphisms of the ACE2 gene (Table 3). The most significant correlation was found for variant ACE2 rs2285666 (AA allele, OR = 2.12, *p* = 0.019) and ACE2 rs2074192 (TT allele, OR = 2.05, *p* = 0.002), and for ACE2 rs4646174 (GG allele, OR = 1.93, *p* = 0.002), ACE2 rs4646156 (TT allele OR = 1.71, *p* = 0.008) and ACE2 rs2158083 (TT allele OR = 1.84, *p* = 0.003).

### 3.2. Association between ACE2 Gene Polymorphisms to General Clinical Characteristics and COVID-19 Disease Severity Outcome

We analyzed the correlation between ACE2 polymorphisms and different demographic and clinical characteristics in study subjects. We found a statistically significant correlation between two ACE2 polymorphisms: rs4646156 (TT > AT, *p* = 0.033025) and rs2158083 (TT > TC, *p* = 0.029705), and higher BMI value. There was no significant correlation between ACE2 polymorphisms and age, diabetes mellitus type 1 or type 2, hypertension, ischemic heart disease and other comorbidities.

## 4. Discussion

The aim of this prospective observational study was to report data regarding the possible association between ACE2 gene polymorphisms and disease severity in patients with SARS-CoV-2 infection. The results of our study demonstrate a statistically significant correlation between the ACE2 receptor gene rs2074192, rs2158083, rs2285666, rs4646156, rs4646174 polymorphisms, and the severity of COVID-19 in adult patients. To our best knowledge, we are the first authors to investigate these five ACE2 polymorphisms together in a wet lab fashion. The data found in other studies vary between these ACE2 gene variants. In our study, we found a strong correlation between ACE2 rs2074192 TT-genotype and poor outcomes in patients with the severe form of COVID-19. Our findings are consistent with a pilot study by Cafiero et al., who identified a higher frequency of T-allele of ACE2 rs2074192 in symptomatic vs. asymptomatic Italian patients [29]. Additionally, a recent study that included 1644 French-Canadian and British patients showed that the T allele was associated with the severity of COVID-19 disease in obese smoking males [30]. Moreover, the rs2074192 T-allele is well known for its relation to cardiovascular risk and hypertension as common risk factors for COVID-19 disease [31]. In a study by Möhlendick et al., the authors postulated that ACE2 rs2285666 GG-genotype is associated with severe COVID-19 outcome, whereas AA-genotype could have a “protective” role [32]. In contrast, we found the opposite conclusions in our Polish population, as rs2285666 AA-genotype increases the risk of being in the severe group of patients, while GA-genotype could be a “protective” variant. This finding could be related to the difference in sample size and the difference in the genetic background of the population. Furthermore, our results contradict two different studies performed recently. In the first study, Gomez et al. compared 204 controls with 204 COVID-19 patients (137 non-severe and 67 severe) and reported no association between ACE2 rs2285666 polymorphism and COVID-19 severity in the Spanish population [33]. However, they found that the ACE1 DD genotype, which was not investigated in our study, together with hypertension, hypercholesterolemia, and hypertensive male gender, were associated with poor outcomes of COVID-19 disease. In the second study, Çelik et al. compared 155 COVID-19 patients via severity of the disease and found no relation between ACE2 rs2106809 and rs2285666 variants with the outcome of COVID-19 disease. In contrast to the results of both above-mentioned studies, our patient groups were more homogenous due to strict exclusion criteria, e.g., older age, severe obesity, and comorbidities, which were intensively used to optimize our findings. Our study revealed the correlation of rs4646156 and rs2158083 with higher BMI values. The ACE2 rs4646156 variant is known to be associated with diabetes mellitus type 2 and high total cholesterol (TC) levels, which are well known to be more frequent in obese patients [20,34]. The severity of SARS-CoV-2 infection is well known to be associated with age, male sex, and some pre-existing diseases, including diabetes mellitus, hypertension, and other cardiovascular diseases. However, we found no significant correlation between ACE2 polymorphisms and age, diabetes mellitus type 1 or type 2, hypertension, ischemic heart disease, and other comorbidities. Sze et al. performed a meta-analysis, which included 18,728,893 patients from 50 studies, and found that individuals from Asian and Black ethnicities had a higher risk of SARS-CoV-2 infection than White individuals [35]. Also, both the rate of ICU admission and death rate were higher in the Asian population. Although our study involved a Polish population of the study subjects, further studies comparing different ethnicities would be required.

The research, however, is subject to several limitations, including a limited sample size. Further research on a larger group of patients is essential for validating our results. The second limitation of concern is that the study subjects were limited to ethnicity and race of the Polish population. A future study in a more diverse patient population might explain some differences in COVID-19 among different ethnic, racial, or geographic backgrounds. However, despite its limitations, this study sheds an important light on the association between the ACE2 gene polymorphisms and the severity of COVID-19.

## 5. Conclusions

In conclusion, we demonstrated that selected ACE2 gene polymorphisms (rs2074192, rs2158083, rs2285666, rs4646156, and rs4646174) are associated with the severity of COVID-19 disease. These five SNPs might be potential genetic candidates for predicting the course of the disease and possibly influencing the treatment choice for COVID-19 disease. Further research involving a larger group of patients should be performed to better understand the influence of genetic variants of the ACE2 gene on COVID-19 severity.

## Figures and Tables

**Table 1 ijerph-19-12622-t001:** Allele frequencies of selected ACE2 gene polymorphisms in the European population.

SNPs	Reference Allele	Alternative Allele
rs2074192	C = 0.569	T = 0.431
rs2158083	C = 0.339	T = 0.661
rs2285666	G = 0.796	A = 0.204
rs4646156	A = 0.408	T = 0.592
rs4646174	C = 0.400	G = 0.600

Legend: ACE2—angiotensin-converting enzyme 2; SNP—single nucleotide polymorphisms.

**Table 2 ijerph-19-12622-t002:** General demographic and clinical characteristics.

Characteristics	Group I	Group II	OR (95% CI)	*p*-Value
Age (mean ± SD)	54.17 ± 3.86	53.07 ± 10.07	NA	0.340
Female, *n* (%)	69 (68.31)	24 (27.59)	0.42 [0.307–0.576]	<0.001
Male, *n* (%)	32 (31.68)	63 (72.41)	2.379 [1.736–3.260]	<0.001
BMI (mean ± SD)	26.10 ± 4.44	28.23 ± 3.97	NA	<0.001
Hypertension, *n* (%)	33 (17.65)	36 (19.25)	1.22 [0.90–1.64]	0.195
Diabetes mellitus type 2, *n* (%)	4 (2.13)	12 (6.38)	3.072 [1.037–9.102]	0.043
Hypothyroidism *n* (%)	16 (15.8)	4 (4.56)	0.208 [0.068–0.638]	0.006

Legend: OR—odds ratio; Cl—confidence interval; NA—not applicable.

**Table 3 ijerph-19-12622-t003:** ACE2 polymorphism genotypes.

ACE2 Polymorphism	Group I	Group II	OR (95% CI)	*p*-Value
rs2285666				
AA	8 (4.26)	16 (8.51)	2.12 [1.13–3.95]	0.019
GG	71 (37.70)	60 (31.91)	0.89 [0.58–1.38]	0.029
GA	22 (11.70)	11 (5.85)	0.53 [0.30–0.94]	0.614
rs4646174				
GG	44 (23.40)	57 (30.32)	1.93 [1.28–2.90]	0.002
CG	31 (16.49)	9 (4.79)	0.43 [0.25–0.75]	0.03
CC	26 (13.83)	21 (11.17)	1.20 [0.75–1.93]	0.449
rs4646156				
TT	42 (22.34)	53 (28.19)	1.71 [1.15–2.53]	0.008
AT	34 (18.09)	13 (6.91)	0.52 [0.32–0.84]	0.008
AA	25 (13.30)	21 (11.17)	1.14 [0.72–1.80]	0.591
rs2158083				
TT	39 (20.74)	53 (28.19)	1.84 [1.24–2.73]	0.003
TC	35 (18.62)	14 (7.45)	0.54 [0.34–0.87]	0.012
CC	27 (14.36)	20 (10.64)	1.00 [0.63–1.59]	0.989
rs2074192				
TT	20 (10.64)	34 (18.09)	2.05 [1.31–3.20]	0.002
CT	36 (19.15)	14 (7.45)	0.47 [0.29–0.75]	0.002
CC	45 (23.94)	39 (20.74)	1.04 [0.70–1.55]	0.834

Legend: OR—odds ratio; Cl—confidence interval.

## Data Availability

Not applicable.

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
