# Peer review of "Association of ACE2 Gene Variants with the Severity of COVID-19 Disease—A Prospective Observational Study"

_ijerph, 2022, doi:10.3390/ijerph191912622_

Round 1

Reviewer 1 Report

1.      Writing language needs to be improved.

2.      Keywords are not standard. Please match the keywords with the mesh.

3.      In the introduction section, the significance of this study need to be strengthens.

4.      Please include limitations section within the discussion section (preferably last paragraph).

5.      Given the nature of the study, it was necessary to seek approval from an ethics committee.

6.      The journal framework is not followed.

Author Response

Dear Reviewer,

On behalf of all the authors, I would like to re-submit the manuscript for your consideration. We carefully read and acknowledge all the comments regarding our manuscript in a detailed point-to-point letter. We hope the manuscript is improved enough to be considered for publication. All authors agreed to the revision and amendments indicated by the reviewers and the authorship has not changed.

Thank you very much for the revision. We appreciate all the comments regarding our manuscript, as they make it stronger.

  1. Comment: “Writing language needs to be improved”

Response: Writing language was corrected and improved in all sections of the manuscripts.

  1. Comment: “Keywords are not standard. Please match the keywords with the mesh”

Response: We reviewed the keywords and adjust them with the mesh. The following corrections were made: “angiotensin-converting enzyme” and “ACE2” were changed to “angiotensin receptors”; COVID-19 was changed to Coronavirus; we added additional keyword that is “single nucleotide polymorphism”

  1. Comment: “In the introduction section, the significance of this study needs to be strengthened”

Response:

  1. Comment: “Please include a limitation section within the discussion section (preferably the last paragraph)

Response: The limitation section was included in the discussion section as the last paragraph.

“The research, however, is subject to several limitations. The first is a sample size that is limited, and further research on a larger group of patients is essential for validating our results. The second limitation concerns, the study subjects were limited to ethnicity and race of the Polish population. A future study in a more diverse patient population might explain some differences during COVID-19 in different ethnic, racial, or geographic backgrounds. However, despite its limitations, this study sheds an important light on the association between the ACE2 gene polymorphisms and the severity of COVID-19. “    

  1. Comment: “Given the nature of the study, it was necessary to seek approval from an ethics committee”

Response: The study was approved by Ethics Committee, and it was included in both the Material and Methods section and Institutional Review Board Statement.

“All experimental protocols were approved by the Ethical Committee of the Pomeranian Medical University in Szczecin (Approval number: KB-0012/88/2020; approval date, June 22.06.2020). The study was conducted in accordance with the Helsinki Declaration (1975, revised 2000). We obtained approval from the Ethical Committee of the Pomeranian Medical University in Szczecin to perform our study without a written informed consent from all study subjects due to the observational nature of the study (Approval number KB-0012/88/2020; approval date, 22.03.2021).”

  1. Comment: “The journal framework is not followed”

Response: Our manuscript was adjusted with the journal framework.

With best regards

Katarzyna Kotfis

Reviewer 2 Report

1- The reason for the selection of five studied SNPs and their potential influence on ACE2 protein must be mentioned in the introduction.

2- References no. 4 and no.16 are less relevant to the manuscript field and can be deleted.

3- How did the researchers check the exclusion criteria such as unknown diabetes or hypertension or decompensated forms of these comorbidities?

4- The "loss of pain" as a COVID-19 infection symptom(lines 70& 113) must be described.

5-How did ICU patients with mechanical ventilation give their informed consent for participation in the research?

6-What is the aim of the sentences at the end of the first paragraph of the results(lines 122-124): "This section may be divided by subheadings. It should provide a concise and precise description of the experimental results, their interpretation as well as the experimental conclusions that can be drawn"?!

7- As the author mentioned: "there was a strong statistical significance between type 2 diabetes mellitus and severe SARS-CoV-2  infection outcome (OR=3.072, p=0.0428)". Thus diabetes is a confounder in group II(Sever form of COVID-19 infection) and can influence the main outcome(disease severity) of this study. How researcher adjusted its effect?

Author Response

Dear Reviewer,

On behalf of all the authors, I would like to re-submit the manuscript for your consideration. We carefully read and acknowledge all the comments regarding our manuscript in a detailed point-to-point letter. We hope the manuscript is improved enough to be considered for publication. All authors agreed to the revision and amendments indicated by the reviewers and the authorship has not changed.

Thank you very much for the revision. We appreciate all the comments regarding our manuscript, as they make it stronger.

  1. Comment: “The reason for the selection of five studied SNPs and their potential influence on ACE2 protein must be mentioned in the introduction.

Response: The reason for the selection of five studied SNPs and their potential influence on ACE2 protein was revised and included in the introduction section.

“In the study, we selected five single nucleotide polymorphisms (SNPs) (rs2074192, rs2158083, rs2285666, rs4646156, rs4646174) based on the frequencies of the variants in the European population. Allele frequencies of these polymorphisms in the European population are presented in table 1. Moreover, several studies have indicated that those SNPs might be involved in SARS-CoV-2 infection. The rs228666 , located in intron 3, may change the mRNA splicing and affect the expression of the ACE2 gene causing greater affinity to a novel coronavirus. Similar properties were found for rs2158083 intronic variant. The rs2074192, an intronic variant, was found to induce RNA changes of secondary structure that may cause ACE2 transcription/translation imbalance. This dysregulation changes SARS-CoV-2 binding to angiotensin receptors [26]. The rs4646156 and rs4646174 were highly associated with left ventricular hypertrophy and primary hypertension, respectively. Cardiovascular diseases are well-known common co-morbidities of SARS-CoV-2 infection that increase the risk of hospitalization and death.”

  1. Comment: “References no. 4 and no. 16 are less relevant to the manuscript field and can be deleted”

Response: Both references no. 4 (Corman, V.M, et al. Hosts and Sources of Endemic Human Coronaviruses) and no. 16 (Voors, A. et al. Dual pathway for angiotensin II formation in human internal mammary arteries.) were deleted from the reference section.

  1. Comment: The “loss of pain” as a COVID-19 infection symptoms (line 70 & 113) must be described.

Response: The loss of pain was mistakenly inserted in the sentence. The correct word was “the loss of smell”. The corrections were included in the section.

  1. Comment: “How did the researchers check the exclusion criteria such as unknown diabetes or hypertension or decompensated forms of these comorbidities?”

Response: The exclusion criteria such as unknown diabetes or hypertension or decompensated forms of these comorbidities were checked by reviewing the medical chart of each patient. The unknown hypertension was established as SBP above 140 mmHg or DBP above 90 mmHg in at least two different measurements with no previous history of hypertension or decompensated hypertension established as hypertension crisis SBP above 180 mmHg and/or DBP above 120 mmHg. The unknown diabetes was established as the random glucose level above 200 mg/dl with no previous history of diabetes, whereas decompensated diabetes was established as diabetic ketoacidosis on admission. Both responses were added to the materials and methods section.

  1. Comment: “How did ICU patients with mechanical ventilation give their informed consent for participation in the research?”

Response: We obtained approval from the Ethical Committee of the Pomeranian Medical University in Szczecin to perform our study without a written informed consent from all study subjects due to the observational nature of the study (Approval number KB-0012/88/2020; approval date, 22.03.2021). The comment was included in the materials and methods section.

  1. Comment: What is the aim of the sentences at the end of the first paragraph of the results (lines 122-124): "This section may be divided by subheadings. It should provide a concise and precise description of the experimental results, their interpretation as well as the experimental conclusions that can be drawn"?!

Response: The sentence was mistakenly inserted in the result section. The sentence was deleted.

  1. Comment: As the author mentioned: "there was a strong statistical significance between type 2 diabetes mellitus and severe SARS-CoV-2 infection outcome (OR=3.072, p=0.0428)". Thus diabetes is a confounder in group II (Sever form of COVID-19 infection) and can influence the main outcome (disease severity) of this study. How researcher adjusted its effect?

Response: Odds ratios and confidence intervals for polymorphisms and concomitant diseases were calculated using the creator of logistic regression. They all were analyzed as one-factor risk evaluation and in conjunction as a multifactor set. Both diabetes and gene polymorphisms were analyzed by using the logistic regression and it was found that diabetes and ACE2 polymorphisms are independent factors for severity of COVID-19. The response was added in the result section of the manuscript.

“Additionally, there was a strong statistical significance between type 2 diabetes mellitus and severe SARS-CoV-2 infection outcome (OR=3.072, p=0.0428).  However, there was no statistical significance between ACE2 gene polymorphisms and diabetes on the severity of COVID-19. Diabetes and ACE2 gene variants seem to be two independent factors of the novel coronavirus outcomes.”

With best regards

Katarzyna Kotfis

Reviewer 3 Report

The authors have evaluated the influence of candidate ACE2 gene polymorphisms on COVID-19 disease risk and severity. I think the preliminary results may be of general scientific interest, although not particularly innovative.

In my opinion, important changes are needed before publishing the paper.

Sample size is limited, and further research on independent replication cohort or in a larger group of patients is necessary to give more value to the data.

There is no multiple testing correction. Please try to apply Bonferroni correction or a similar measure (FDR, BDFP), in order to account for multiple testing.

Although the authors reported same significant associations, the authors did not suggest any hypothesis  of link between ACE2 variants and severity of the disease. The authors may consider changing the discussion.

The authors should also provide the genotype frequencies of these polymorphisms in the general population. It would also be advisable to calculate the Hardy-Weinberg equilibrium of each SNPs studied.

How were the SNPs selected? And where are the they located? (Intronic, exonic, splicing site, etc ...)

Some results observed in this study are opposite to those reported in the literature (eg line 158). How do the authors explain this?

Author Response

Dear Reviewer,

On behalf of all the authors, I would like to re-submit the manuscript for your consideration. We carefully read and acknowledge all the comments regarding our manuscript in a detailed point-to-point letter. We hope the manuscript is improved enough to be considered for publication. All authors agreed to the revision and amendments indicated by the reviewers and the authorship has not changed.

Thank you very much for the revision. We appreciate all the comments regarding our manuscript, as they make it stronger.

  1. Comment: “Sample size is limited, and further research on independent replication cohort or in a larger group of patients is necessary to give more value to the data.”

Response: Authors mentioned the sample size in the limitation in the discussion section.

“The research, however, is subject to several limitations. The first is a sample size that is limited, and further research in a larger group of patients is essential for validating our results. The second limitation concerns, the study subjects were limited to ethnicity and race of the Polish population. A future study in a more diverse patient population might explain some differences during COVID-19 in different ethnic, racial or geographic backgrounds. However, despite its limitations this study sheds an important light onto the association between the ACE2 gene polymorphisms and the severity of COVID-19. “   

  1. Comment: “There is no multiple testing correction. Please try to apply Bonferroni correction or a similar measure (FDR, BDFP), to account for multiple testing.”

Response: In our study we calculated odds ratios and confidence intervals for polymorphisms and concomitant diseases were using the creator of logistic regression. They all were analyzed as one-factor risk evaluation and in conjunction as a multifactor set. In our study, a dichotomous dependent variable (disease severity) and a nominal independent variable (polymorphism) were analyzed. In most studies, the Bonferron’s correction was applied to multiple assessments of the compared means, which was not the case of our study. However, in the article (Pernerg T What’s wrong with Bonferroni adjustment”) it was mentioned that it is used in cases like ours – only that the author questions the sense of such an action.  “The paper advances the view widely held by epidemiologists that Bonferroni adjustments are, at best, unnecessary and, at worst, deleterious to sound statistical inference”.

  1. Comment: “Although the authors reported same significant associations, the authors did not suggest any hypothesis of link between ACE2 variants and severity of the disease.”

Response: The hypothesis of link between ACE2 variants and severity of the disease was based on the influence of those five SNPs on ACE2 and simultaneously the ability of the SARS-CoV-2 virus to bind the ACE2 receptor that might affect the severity of the SARS-Co2-2 infection. The introduction and discussion sections were adjusted accordingly.  

  1. Comment: “The authors should also provide the genotype frequencies of these polymorphisms in the general population. It would also be advisable to calculate the Hardy-Weinberg equilibrium of each SNPs studied.”

Response: The genotype frequencies of selected polymorphisms in the European population are presented in the table 1 that was added to the introduction section. Table 2 presents the Hardy-Weinberg equilibrium of each studied SNPs.

Table 1. Allele frequencies of selected ACE2 gene polymorphisms in the European population

SNPs

Reference allele

Alternative allele

rs2074192

C=0.569

T=0.431

rs2158083

rs2285666

rs4646156

rs4646174

C=0.339

G=0.796

A=0.408

C=0.400

T=0.661

A=0.204

T=0.592

G=0.600

 Legend: ACE2 – angiotensin-converting enzyme 2; SNP – single nucleotide polymorphisms

Table 2. The Hardy-Weiberg equilibrium of each SNPs

Genotype

rs2285666

GroupI[n(%)]

GroupII[n(%)]

Observed values

n (%)

Expected values

%

Observed values

n (%)

Expected values

%

OR

95% CI

P

GG

71(70.30)

65.92

60(68.97)

56.69

0.89[0.58-1.38]

0.029

GA

22(21.78)

30.54

11(12.64)

37.20

0.53[0.30-0.94]

0.614

AA

8(7.92)

3.54

16(18.39)

6.11

2.12[1.13-3.95]

0.019

Total

101 (100)

100

87 (100)

100

-

-

Alleles

G

164 (81.19)

-

131 (75.29)

-

A

38 (18.81)

-

43 (24.71)

-

Total

202 (100)

-

174 (100)

-

-

-

Genotype

rs4646174

GroupI[n(%)]

GroupII[n(%)]

Observed values

n (%)

Expected values

%

Observed values

n (%)

Expected values

%

OR

95% CI

P

GG

44(43.56)

33.64

57(65.52)

49.97

CG

31(30.69)

49.48

9(10.34)

41.44

CC

26(25.75)

16.88

21(24.38)

8.59

Total

101 (100)

100

87 (100)

100

-

-

Alleles

G

119 (58.91)

-

123 (70.69)

-

C

83 (41.09)

-

51 (29.31)

-

Total

202 (100)

-

174 (100)

-

-

-

Genotype

rs4646156

GroupI[n(%)]

GroupII[n(%)]

Observed values

n (%)

Expected values

%

Observed values

n (%)

Expected values

%

OR

95% CI

P

TT

42(41.58)

34.12

53(60.92)

46.77

AT

34(33.67)

48.59

13(14.94)

43.24

AA

25(24.75)

17.29

21(24.14)

9.99

Total

101 (100)

100

87 (100)

100

-

-

Alleles

T

118 (58.42)

-

119 (68.39)

-

A

84 (41.58)

-

55 (31.61)

-

Total

202 (100)

-

174 (100)

-

-

-

Genotype

rs2158083

GroupI[n(%)]

GroupII[n(%)]

Observed values

n (%)

Expected values

%

Observed values

n (%)

Expected values

%

OR

95% CI

P

TT

39(38.61)

31.29

53(60.92)

47.57

TC

35(34.65)

49.30

14(16.09)

42.80

CC

27(26.74)

19.41

20(22.99)

9.63

Total

101 (100)

100

87 (100)

100

-

-

Alleles

T

113 (55.94)

-

120 (68.97)

-

C

89 (44.06)

-

54 (31.03)

-

Total

202 (100)

-

174 (100)

-

-

-

Genotype

rs2074192

GroupI[n(%)]

GroupII[n(%)]

Observed values

n (%)

Expected values

%

Observed values

n (%)

Expected values

%

OR

95% CI

P

TT

20(19.80)

14.15

34(39.08)

22.21

CT

36(35.64)

46.94

14(16.09)

49.84

CC

45(44.56)

38.91

39(44.83)

27.95

Total

101 (100)

100

87 (100)

100

-

-

Alleles

T

76 (37.62)

-

82 (47.13)

-

C

126 (62.38)

-

92 (52.87)

-

Total

202 (100)

-

174 (100)

-

-

-

  1. Comment: “How were the SNPs selected? And where are they located? (Intronic, exonic, splicing site, etc ...)”

Response: The SNPs were selected based on the frequency of these polymorphisms in the European population. The five selected SNPs were located in the introns. Moreover, many studies described the important role of selected SNPs in SARS-CoV-2 infections. The findings were added in the introduction section.

“In the study, we selected five single nucleotide polymorphisms (SNPs) (rs2074192, rs2158083, rs2285666, rs4646156, rs4646174) based on the frequencies of the variants in the European population. Allele frequencies of these polymorphisms in the European population are presented in table 1. Moreover, several studies have indicated that those SNPs might be involved in SARS-CoV-2 infection. The rs228666, located in intron 3, may change the mRNA splicing and affect the expression of the ACE2 gene causing greater affinity to a novel coronavirus. Similar properties were found for rs2158083 intronic variant [25, 26]. The rs2074192, an intronic variant, was found to induce RNA changes of secondary structure that may cause ACE2 transcription/translation imbalance. This dysregulation changes SARS-CoV-2 binding to angiotensin receptors. The rs4646156 and rs4646174 were highly associated with left ventricular hypertrophy and primary hypertension, respectively. Cardiovascular diseases are well-known common co-morbidities of SARS-CoV-2 infection that increase the risk of hospitalization and death.”

  1. Comment: “Some results observed in this study are opposite to those reported in the literature (eg. line 158). How do the authors explain this?”

Response: This finding could be related to difference in sample size and the difference in the genetic background between the population. The statement was added to the discussion selection.

“In a study by Möhlendick et al., the authors postulated that ACE2 rs2285666 GG-genotype is associated with severe COVID-19 outcomes, whereas AA-genotype could have a “protective” role [32]. In contrast, we found opposite conclusions in our Polish population as rs2285666 AA-genotype increases the risk of being in the severe group of patients, and GA-genotype could be a “protective” variant. This finding could be related to the difference in sample size and the difference in the genetic background between the population.”

With best regards

Katarzyna Kotfis

Round 2

Reviewer 1 Report

All suggested comments have been implemented in the manuscript.

Reviewer 2 Report

Thank you for revising the manuscript.

Reviewer 3 Report

The authors replied to all previous comments